# Correlation between baseline anion gap and early acute kidney injury in patients with acute pancreatitis in the intensive care unit: A single-center retrospective cohort study

**Yaqing Zhang[1], Haiping Ma[1], Rui Wang[2], Li Li[1], Qingwei Kong[1], Cuiping Hao[2], Ying Zhang[1], Jinfeng Li [3]***

1 Department of Hepatobiliary Surgery, Affiliated Hospital of Jining Medical University, Jining, Shandong, China, 2 Department of Critical Care Medicine, Affiliated Hospital of Jining Medical University, Jining, Shandong, China, 3 Department of Gastrointestinal Surgery, Affiliated Hospital of Jining Medical University, Jining, Shandong, China

* wcwkljf@163.com

## Abstract

### Objective

Acute pancreatitis (AP) is an acute inflammatory disease that can lead to multiple system dysfunction, including acute kidney injury (AKI). AKI occurs in 10%-42% of AP patients, and studies have shown that early (48 hours) acute pancreatitis associated acute kidney injury (AP-AKI) can increases the risk of death in acute pancreatitis. Anion gap (AG) is a common index in clinical evaluation of acid-base imbalance and an important index in critically ill patients. The aim of this study was to investigate the relationship between baseline anion gap values and early acute kidney injury in patients with acute pancreatitis in intensive care unit.

### Methods

Our data were derived from inpatients admitted to Beth Israel Deaconess Medical Center (BIDMC) in the United States between 2008 and 2019. A total of 4,017 adult patients with acute pancreatitis admitted to the ICU were enrolled in the study, and 475 were enrolled according to the exclusion and inclusion criteria. Only the baseline value and one day after arrival to the intensive care unit (ICU) were considered for all laboratory test values. According to previous literature and clinical significance, AG was divided into two groups: low value (< 16mmol/L) group and high value (≥16mmol/L) group, and logistics univariate and multifactor regression analysis was applied to verify the relationship between anion gap and AKI risk.

### Results

Only 157 of the 475 AP cases had an AG level below 16 mmol/L, whereas 318 patients had an AG level over 16 mmol/L. Within 48 hours, 89 and 240 cases (56.7% and 75.5%) and the low- and high-AG groups had AKI. In AP cases, an elevated AG was related to an increased risk of AKI [odds ratio (OR) = 1.06, 95% confidence interval (CI): 1.03–1.1], and is a

**Data Availability Statement:** All data for this study are available in the dryad database (DOI: 10.5061/

dryad.tdz08kQ72) and in the Supporting
Information File (S1 File).

**Funding:** The author(s) received no specific
funding for this work.

**Competing interests:** The authors have declared
that no competing interests exist.

nonlinear relationship. When controlling for other factors, this correlation was still significant.
Compared to the lower group, high-AG ($\geq$16mmol/L) values can increase the risk of early
acute kidney injury in patients with acute pancreatitis (OR = 2.35, CI: 1.57–3.53).

## Conclusion

Anion gap (AG) is an independent risk factor for early acute kidney injury in patients with
acute pancreatitis, and has a nonlinear relationship with 48-hour AKI. Higher AG($\geq$16mmol/
L) values can significantly increase the risk of AP-AKI.

## Introduction

Acute pancreatitis (AP) is a pathological condition characterized by inflammation of the exo-
crine pancreas, which is accompanied by tissue damage and necrosis [1]. The number of hos-
pitalizations for AP cases has shown a consistent upward trend in both Europe and the United
States during the last decade [2]. Around 20% of patients experience the development of mod-
erate or severe AP, characterized by necrosis of the pancreatic or peripancreatic tissue, organ
failure [2,3], and are associated with an increased morbidity and mortality [3]. During the first
stages of AP, the activation of inflammation inside the inflamed pancreas initiates vascular
damage, which subsequently leads to endothelial dysfunction, activation of coagulation,
heightened recruitment of leukocytes, and their subsequent migration to the pancreas, con-
tributing to the progression of AP [4–7]. Among the organ injuries caused by AP, there is an
incidence of acute kidney injury (AKI) seen in around 10% to 42% of cases [7–11].

AKI is a prevalent condition, particularly seen in individuals with severe illness [12,13]. The
present diagnosis of AKI relies on the abrupt decline in renal function, as shown by elevated
levels of creatinine and/or reduced urine output [14]. Nevertheless, the description of this syn-
drome comprises a diverse range of clinical symptoms, varying pathophysiology, etiology, and
risk factors, as well as distinct short-and long-term outcomes [15–17]. For many decades, med-
ical professionals have been aware that AKI is a potentially fatal consequence of AP. However,
the degree to which this complication contributes to patient mortality is frequently not fully
recognized [18–20]. Previous retrospective investigations identified many risk factors for AKI
in cases of AP. These risk factors included a history of renal disease, hypoxemia, and acid-base
imbalance. Additionally, intensive care unit (ICU) admission, respiratory failure, sepsis, male
gender, older age, and a history of chronic kidney disease (CKD) were shown to be associated
with an increased risk of AKI [17–19,21]. Researchers are also endeavoring to ascertain prog-
nostic indicators in AKI. Regrettably, the majority of these biomarkers are not extensively
used in clinical settings [9,21–23]. Moreover, there exists a dearth of robust data to substantiate
the efficacy of any particular biomarker for the timely identification of AKI [24].

An acid-base balance is the basis for maintaining cell metabolism and physiological func-
tion [25]. The anion gap (AG) is a longstanding clinical indication that is used to assess acid-
base balance in conventional clinical settings. Its usage in clinical practice has spanned over
five decades [26,27]. The determination of the serum AG has been used to identify inaccura-
cies in the measurement of serum electrolytes, identify paraproteins in blood, and, notably for
nephrologists, identify and assess metabolic acidosis [27]. The baseline AG in adult patients in
the ICU has been reported to be a sensitive and specific tool for predicting prognosis or death
[25,28]. Nevertheless, there is a lack of extensive research on the correlation between baseline

AG and AKI in AP cases [9,20,22,26,29]. This research will examine the correlation between baseline AG and early(48 hours)AKI occurring in individuals with AP.

## Materials and methods

### Database

The data utilized in this work were exclusively sourced from the MIMIC-IV (v2.2) database. This database, established and overseen by the MIT Computational Physiology Laboratory, is a substantial and openly available resource (https://physionet.org/content/mimiciv/1.0/) [30,31].The database has comprehensive information about all patients who were admitted to the Beth Israel Deaconess Medical Center (BIDMC) between the years 2008 and 2019 [30–32]. To safe guard patient confidentiality, personal information was anonymized by the utilization of random numbers in lieu of patient identification. Consequently, the acquisition of informed permission and ethical clearance from patients was not required [32]. To get access to the database, Yaqing Zhang, the primary author of this work, successfully completed the Collaborative Institutional Training Initiative (CITI) course and achieved passing scores on both the "Data or Specimens Only Research" and "Conflicts of Interest" examinations (ID: 12197648). The study team obtained the necessary qualifications to access the database and get data.

### Study population

The hospitalization data of AP cases was collected using the 10th and 9th Revision of the International Classification of Diseases (ICD-10 and -9) code K85-K8592 and 5770, respectively. The study enrolled a total of 4,017 patients who were admitted to the intensive care unit (ICU), with all participants being 18 years of age or older. According to the exclusion criteria:1. Not the first time admitted to ICU; 2. Stay in ICU for less than 24 hours;3. With covariate data missing>30%, a total of 475 cases were included in the study population (Fig 1).

### Definition of AG and diagnosis of AKI

The AG refers to the numerical disparity between the total concentration of cations seen in a given sample and that anions present. This study included the extraction of baseline AG values of AP patients after admission to ICU using Structured Query Language (SQL) in Navicat.The chosen outcome measure was the incidence of AKI within 48 years in patients with acute pancreatitis.The diagnosis of AKI was established according to the 2012KDIGO (Organization for Improving Global Outcomes in Kidney Disease) Guidelines for AKI. This diagnostic criterion is an rise in serum creatinine (SCr) levels of 26.5 lmol/L (0.3 mg/dl) over a 48-hour period.

### Covariate selection

The covariates in this study refer to previous research reports on the one hand, and to clinical study significance on the other hand. These variables are commonly used in studies of the relationship between AG and AKI, and may reflect disease severity and risk of AKI in AP subjects. The variables to be included in the study encompass demographic characteristics such as age, sex, and ethnicity. Additionally, the length of hospital and ICU stays, and mortality will be considered. Furthermore, vital signs, laboratory test results, specifically hemoglobin (HB), white blood cell count (WBC), blood urea nitrogen (Bun), lactic acid(LAC), platelet count (PLT), serum albumin, and bicarbonate (HCO3-) gathered within the 1st day of ICU admission were analyzed. Simplified Acute Physiology Score (SAPS) II and Sequential Organ Failure Assessment (SOFA) are also be select.

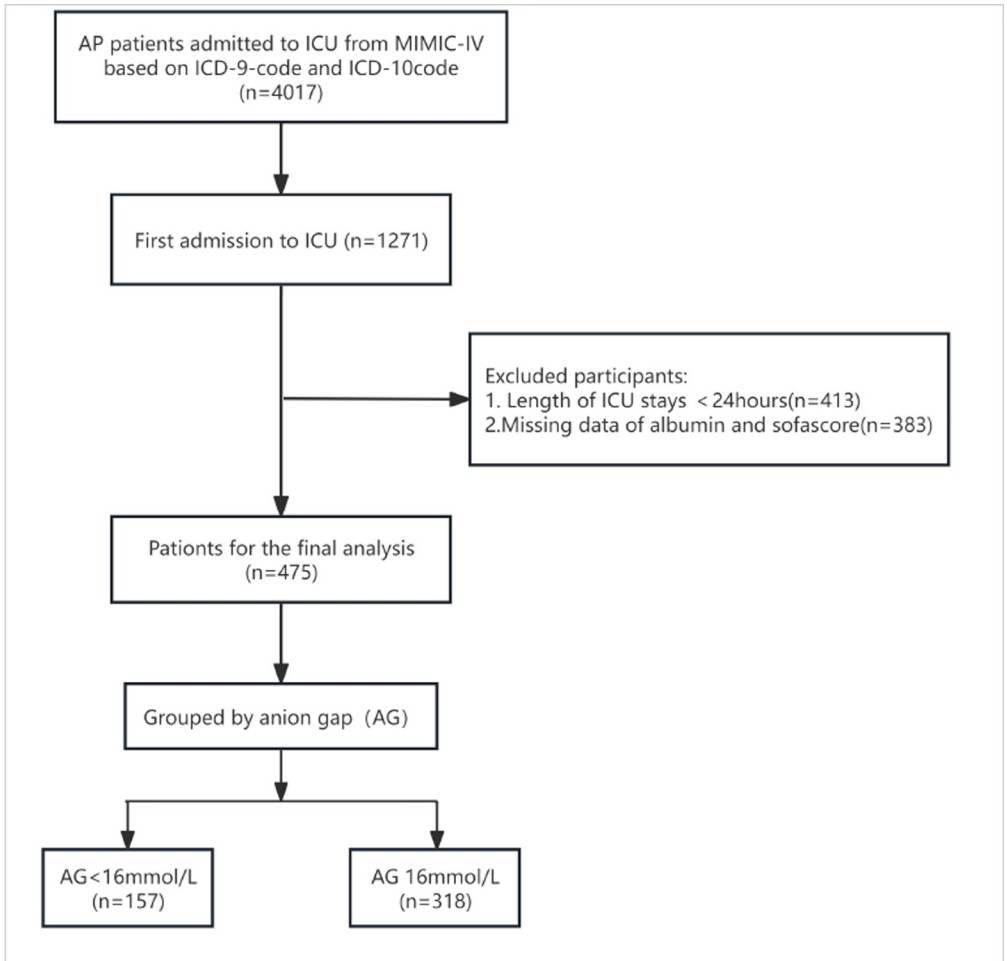

**Fig 1. Flowchart exhibiting the selection process.**

## Statistical analysis

The Shapiro-Wilk statistical test was employed for assessing the normality of a continuous variable's distribution. The categorical variables were represented as percentages, and $\chi2$ statistics were utilized for the sake of comparison. The continuous variables that followed a normal distribution could be characterized by their means and standard deviation (SD), while the skewed variables were defined by their median and interquartile range (IQR). The statistical analysis of the data included the utilization of the Kruskal-Wallis H test to assess the findings. In order to minimize confounding, we directly exclude data with covariate missingness $\geq$10%, and for variables with <10% missingness, we employ multiple imputation based on the 5 replications method in the Free statistical program to account for the missing data of the covariate.The odds ratios (OR) and corresponding 95% confidence intervals (CI) were computed using logistic univariate and multifactorial regression techniques to evaluate the correlation between baseline AG and AKI incidence. The variable AG was converted into binary variables, namely the high value group (AG $\geq$16mmol/L) and the low value group (AG < 16mmol/L). Regression analysis was conducted on a combination of continuous and categorical data. The study included several models: an unadjusted model, an original model, and Model 1, which was adjusted for age, sex, and race. In the second model, an increase in oxygen saturation,

respiratory and heart rates, and blood pressure were observed compared to the first model. The third model, building upon the second model,plus lactic acid and sofa score.A limited cubic spline was employed for assessing linearity and provide deeper insight into the nature of the dose-response correlation between AG and AKI. The smooth curve fitting plot was constructed and modified, and the analysis of inflection points was conducted based on the adjusted variables of model 3.

The statistical software packages Free Statistics software version 1.9 were employed for all studies. A comprehensive investigation was undertaken including all participants in order to gather descriptive data. A p-value less than 0.05 is indicative of statistical significance when using a two-tailed test.

## Results

### Baseline characteristics

Based on the baseline AG within 24 hours of admission to the ICU, the participants were categorized into two groups: the low (AG <16 mmol/L) and high (AG ≥16 mmol/L) value groups (Table 1). The whole population consisted of 202 females and 273 men, with 48.2% of the patient population under 60 years, while the remaining (51.8%) were 60 years and above. Significant variations in serum AG levels were seen between the two groups across many variables, including WBC counts, serum albumin, HCO3- levels, BUN, calcium, chloride, glucose, sodium and potassium levels, and different severity of illness ratings. AKI was seen in 89 and 240 patients (56.7% and 75.5%) in the low- and high-value groups during a 48-hour timeframe (Table 1).

### Univariate analysis

Table 2 presents the findings indicating a positive correlation between many factors, including the duration of hospital and ICU stays, heart and respiration rates, WBC count, serum albumin, HCO3-, lactic acid, BUN, calcium, chloride, creatinine and potassium levels, SapsII scores, Sofa score and the incidence of AP-related AKI. There was a negative correlation between blood pressure and blood oxygen saturation and the incidence of AP-associated AKI.

### Baseline anion gap and acute kidney injury

Multivariate regression evaluated the relationship between AG and AKI (Table 3). The outcomes of the unadjusted model, Models 1, 2, and 3 are shown in Table 3. In the unadjusted model, a positive correlation was seen between baseline AG and AKI, with an OR of 1.06(95% CI:1.03–1.1, P = 0.01). After controlling for confounding factors like race, sex, and age in Model 1, and further correcting for other variables (as previously discussed) in Model 2, a positive correlation between AKI and the aforementioned variables persisted (OR:1.05,95% CI:1.02–1.09, p-value = 0.05). Despite the inclusion of additional adjusted factors (as previously explained), The results of Model 3 also indicate asignificant correlation between the variable of interest and the outcome (OR = 1.05,95%CI:1.01–1.09, p-value = 0.015).

In multivariate regression, the inclusion of serum AG as a categorical variable revealed that the high value group exhibited a 135% higher risk of AKI compared to the low value one in the unadjusted model (OR:2.35,95%CI:1.57–3.53, p<0.01). Following the adjustment of variables in Models 1, 2, and 3, the risk of AKI exhibited corresponding increases of 141% (OR:2.41,95% CI:1.60–3.64), 126% (OR:2.26, 95%CI:1.48–3.45), and 109% (OR:2.09, 95%CI: 1.35–3.24), respectively.

**Table 1. Characteristics of the included patient population.**

| Variables | Total (n = 475) | AG<16 (n = 157) | AG≥16 (n = 318) | p |
|---|---|---|---|---|
| **Demographics** | | | | |
| Age,n(%) | | | | 0.400 |
| <60years | 229 (48.2) | 80 (51) | 149 (46.9) | |
| ≥60years | 246 (51.8) | 77 (49) | 169 (53.1) | |
| Gender,n(%) | | | | 0.659 |
| Female | 202 (42.5) | 69 (43.9) | 133 (41.8) | |
| Male | 273 (57.5) | 88 (56.1) | 185 (58.2) | |
| Race,n(%) | | | | 0.168 |
| White | 286 (60.2) | 104 (66.2) | 182 (57.2) | |
| Black | 40 (8.4) | 11 (7) | 29 (9.1) | |
| Ather | 149 (31.4) | 42 (26.8) | 107 (33.6) | |
| **Vital signs** | | | | |
| Heart rate,beat/min | 115.5 ± 23.2 | 112.9 ± 21.4 | 116.7 ± 24.0 | 0.086 |
| SBP,mmHg | 91.8 ± 19.5 | 93.0 ± 19.5 | 91.2 ± 19.5 | 0.360 |
| DBP,mmHg | 48.2 ± 13.2 | 49.1 ± 12.5 | 47.8 ± 13.5 | 0.293 |
| Resp rate,beat/min | 30.4 ± 7.3 | 30.1 ± 7.3 | 30.6 ± 7.3 | 0.442 |
| Spo2,% | 90.4 ± 6.9 | 91.7 ± 3.9 | 89.8 ± 7.8 | 0.007 |
| **Laboratory index** | | | | |
| Hb,mmol/L | 10.2 ± 2.3 | 10.0 ± 2.0 | 10.3 ± 2.4 | 0.214 |
| PLT,$10^9$/L | 153.0 (103.0, 219.0) | 178.0 (112.0, 237.0) | 140.5 (101.2, 201.8) | < 0.001 |
| WBC,$10^9$/L | 14.9 (10.4, 21.7) | 13.0 (9.9, 18.2) | 15.8 (10.6, 22.3) | 0.002 |
| Albumin,g/L, | 2.9 ± 0.7 | 2.7 ± 0.6 | 3.0 ± 0.7 | < 0.001 |
| $HCO_3^-$,mmol/L | 18.6 ± 5.8 | 22.3 ± 4.4 | 16.8 ± 5.5 | < 0.001 |
| AG,mmol/L | 19.3 ± 6.9 | 13.3 ± 1.5 | 22.2 ± 6.5 | < 0.001 |
| ACAG,mmol/L | 23.0 ± 6.9 | 17.5 ± 1.8 | 25.7 ± 6.8 | < 0.001 |
| Cr,mmol/L | 1.5 (0.9, 2.7) | 1.0 (0.7, 1.3) | 2.0 (1.1, 3.6) | < 0.001 |
| Glucose,mg/dl | 153.0 (115.0, 224.5) | 134.0 (105.0, 169.0) | 172.5 (124.2, 252.8) | < 0.001 |
| LAC,mmol/L | 2.2 (1.4, 3.8) | 1.7 (1.3, 2.9) | 2.5 (1.6, 4.5) | < 0.001 |
| Bun,mmol/L | 28.0 (17.0, 49.0) | 18.0 (14.0, 28.0) | 35.0 (22.0, 56.8) | < 0.001 |
| $Ca^{2+}$,mmol/L | 7.4 ± 1.1 | 7.5 ± 1.0 | 7.4 ± 1.2 | 0.387 |
| $Cl^-$,mmol/L | 102.3 ± 8.0 | 106.1 ± 6.0 | 100.4 ± 8.3 | < 0.001 |
| $Na^+$,mmol/L | 140.8 ± 6.0 | 141.4 ± 4.4 | 140.5 ± 6.6 | 0.107 |
| $K^+$,mmol/L | 4.5 ± 0.8 | 4.2 ± 0.6 | 4.7 ± 0.9 | < 0.001 |
| ALT,U/L | 69.0 (29.0, 199.5) | 45.0 (23.0, 126.0) | 87.0 (34.0, 237.0) | < 0.001 |
| AST,U/L | 106.0 (50.0, 259.0) | 61.0 (35.0, 133.0) | 125.0 (64.0, 341.5) | < 0.001 |
| **Comorbidity,n(%)** | | | | |
| Heartfailure | | | | < 0.001 |
| NO | 386 (81.3) | 143 (91.1) | 243 (76.4) | |
| YES | 89 (18.7) | 14 (8.9) | 75 (23.6) | |
| Lild liver disease | | | | 0.248 |
| NO | 319 (67.2) | 111 (70.7) | 208 (65.4) | |
| YES | 156 (32.8) | 46 (29.3) | 110 (34.6) | |
| Diabetes | | | | 0.015 |
| NO | 458 (96.4) | 156 (99.4) | 302 (95) | |
| YES | 17 (3.6) | 1 (0.6) | 16 (5) | |
| Renal disease | | | | < 0.001 |
| NO | 395 (83.2) | 144 (91.7) | 251 (78.9) | |

*(Continued)*

**Table 1.** (Continued)

| Variables | Total (n = 475) | AG<16 (n = 157) | AG≥16 (n = 318) | p |
|---|---|---|---|---|
| YES | 80 (16.8) | 13 (8.3) | 67 (21.1) | |
| **Critical care score** | | | | |
| SapsII score | 42.5 ± 15.9 | 35.2 ± 13.4 | 46.1 ± 15.9 | < 0.001 |
| Sofa score | 4.4 ± 2.5 | 4.0 ± 2.3 | 4.6 ± 2.5 | 0.014 |
| **Outcomes,n(%)** | | | | |
| AKI-2days | | | | < 0.001 |
| NO | 146 (30.7) | 68 (43.3) | 78 (24.5) | |
| YES | 329 (69.3) | 89 (56.7) | 240 (75.5) | |
| Hospital.stays, Median (IQR) | 15.0 (8.0, 27.0) | 15.0 (9.0, 27.0) | 15.0 (8.0, 27.0) | 0.628 |
| ICU stays(days) | 5.0 (2.0, 13.0) | 5.0 (2.0, 11.0) | 5.0 (2.0, 13.0) | 0.418 |
| Death in-hospital | | | | 0.002 |
| NO | 315 (66.3) | 119 (75.8) | 196 (61.6) | |
| YES | 160 (33.7) | 38 (24.2) | 122 (38.4) | |

SBP:Systolic blood pressure;DBP:Diastolic blood pressure;Spo2:Blood oxygen degree of saturation;Hb:Hemoglobin.

PLT:Blood platelet;WBC:White blood cell count;AG:Anion gap;ACAG:Albumin corrected anion gap;HCO3⁻:Bicarbonate radical.

Bun:Blood urea nitrogen;LAC:Lactic acid;Ca²⁺:Blood calcium;Cl⁻:Chloridion;Cr:Serum creatinine;Na⁺:Sodium;K+:Serum potassium.

ALT:Alanine aminotransferase;AST:Glutamic oxalacetic transaminase;SapsII:Simplified Acute Physiology Score.

Sofa:Sequential Organ Failure Assessment.

## Nonlinear relationship between anion gap and acute kidney injury

Following the adjustment for certain variables, a limited cubic spline model examined the link between baseline AG and AKI, with an S-curve pattern (Fig 2). Logistic curve fitting showed an S-shaped curve relationship between baseline AG and 48h-AKI outcomes in patients with acute pancreatitis. Logistic linear regression model was used to calculate the threshold inflection points of 15mmol/L and 28mmol/L. Between the two inflection points, the number of acute kidney injury increased with the increase of AG value, which was consistent with the results of binary classification of AG (S1 Table).

## Discussion

Acute kidney injury (AKI) is a common complication of various acute diseases, including acute pancreatitis (AP) [2,12,33]. In recent years, people have increasingly paid attention to the relationship between serum anion gap (AG) and the development of AKI [22,29,34]. In this study, we investigated the correlation between baseline AG values and early AKI in critically ill patients with AP.

Consistent with previous studies, our study demonstrated an independent correlation between elevated serum AG levels and increased AKI risk in AP patients. Firstly, after adjusting for age, sex, race, vital signs, bicarbonate, albumin, blood lactic acid and SOFA scores,multifactor regression analysis using AG as a binary variable showed that patients with high (≥16mmol/L)AG value significantly increased the risk of AKI. Secondly, we observed a nonlinear relationship between AG levels and AKI risk, characterized by an s-shaped correlation. Between the two inflection points of 15mmol/L and 28mmol/L, the number of acute kidney injury increased with the increase of AG value. However, this trend was opposite on the right side of the inflection point, and the clinical significance was not significant, which may be related to the small sample size of AG > 28mmol/L. In clinical practice, the results of this

**Table 2. Association between different covariates and Acute Kidney Injury.**

| Variable | OR (95%CI) | P |
|---|---|---|
| Age,n(%) | | |
| <60years | 1(Reference) | |
| ≥60years | 1.15 (0.78～1.7) | 0.472 |
| Gender,n(%) | | |
| Female | 1(Reference) | |
| Male | 1.08 (0.73～1.6) | 0.701 |
| Race,n(%) | | |
| White | 1(Reference) | |
| Black | 1.23 (0.57～2.62) | 0.598 |
| Others | 0.74 (0.49～1.13) | 0.163 |
| Heart rate,beat/min, | 1.01 (1～1.02) | 0.045 |
| SBP,mmHg | 0.98 (0.97～0.99) | <0.001 |
| DBP,mmHg | 0.98 (0.96～1) | 0.011 |
| Resp rate,beat/min | 1.02 (0.99～1.05) | 0.115 |
| Spo2,% | 0.94 (0.9～0.98) | 0.005 |
| HB,mmol/L | 0.97 (0.89～1.06) | 0.491 |
| PLT,$10^9$/L | 1 (1～1) | 0.236 |
| WBC,$10^9$/L | 1.04 (1.01～1.06) | 0.002 |
| Albumin,g/L | 0.56 (0.41～0.75) | <0.001 |
| AG,mmol/L | 1.06 (1.03～1.1) | 0.001 |
| ACAG,mmol/L | 1.09 (1.05～1.13) | <0.001 |
| LAC,mmol/L | 1.1 (1.02～1.19) | 0.017 |
| $HCO3^-$,mmol/L | 0.93 (0.9～0.97) | <0.001 |
| Bun,mmol/L | 1.01 (1.01～1.02) | 0.001 |
| $Ca^{2+}$,mmol/L | 0.64 (0.52～0.78) | <0.001 |
| $Cl^-$,mmol/L | 0.96 (0.94～0.99) | 0.005 |
| Cr,mmol/L | 1.48 (1.26～1.73) | <0.001 |
| Glucose,mg/dl | 1 (1～1) | 0.048 |
| $Na^+$,mmol/L | 0.95 (0.92～0.98) | 0.004 |
| $K^+$,mmol/L | 1.8 (1.36～2.38) | <0.001 |
| ALT,U/L | 1 (1～1) | 0.297 |
| AST,U/L | 1 (1～1) | 0.105 |
| Sofascore | 1.24 (1.12～1.36) | <0.001 |
| SapsII | 1.06 (1.04～1.08) | <0.001 |
| **Comorbidity,n(%)** | | |
| Heartfailure | | |
| NO | 1(Reference) | |
| YES | 1.44 (0.85～2.44) | 0.174 |
| Lild_liver_disease,n(%) | | |
| NO | 1(Reference) | |
| YES | 0.95 (0.63～1.44) | 0.824 |
| Diabetes,n(%) | | |
| NO | 1(Reference) | |
| YES | 2.12 (0.6～7.49) | 0.244 |
| Renal disease,n(%) | | |
| NO | 1(Reference) | |
| YES | 1.52 (0.87～2.65) | 0.139 |

(*Continued*)

**Table 2.** (Continued)

| Variable | OR (95%CI) | P |
|---|---|---|
| Hospital stays(day) | 1.05 (1.03 ~ 1.06) | <0.001 |
| ICU stays(day) | 1.08 (1.04 ~ 1.11) | <0.001 |
| Dod,n(%) | | |
| NO | 1(Reference) | |
| YES | 1.68 (1.09 ~ 2.58) | 0.019 |

Abbreviations: CI,confidence interval; OR, odds ratio.

study can correctly guide the medical staff to timely evaluate the severity of the disease for AP patients newly admitted to ICU, and guide more targeted management strategies.

The underlying mechanisms explaining the correlation between AG and AKI in AP patients are multifactorial and complex. Elevated AG levels may reflect metabolic acidosis and impaired kidney function, both of which are common in AP. In addition, AG is associated with other factors involved in the pathogenesis of AKI, including inflammation, oxidative stress, and endothelial dysfunction. Further research is needed to elucidate the exact mechanism of the observed correlation between AG and AKI in AP [1,12,20,28,35–36].

However, the anion gap is a simple and easily obtained laboratory indicator, which can evaluate the prognosis of patients with acute pancreatitis as early as possible, especially the occurrence of AKI, which can guide further rescue treatment [29,34].

The study conducted possessed several strengths, such as a substantial sample size and the incorporation of adjustments for multiple covariates. These covariates encompassed demographic characteristics, vital signs, laboratory results, and comorbidities, which were accounted for to mitigate potential confounding factors. This approach aimed to minimize confounders and obtain a more accurate picture of the relationship between AG and AKI. Nevertheless, there are still some limitations that we must acknowledge. As a retrospective observational study, our ability to establish a causal relationship between serum anion gap (AG) and acute kidney injury (AKI) in patients with acute pancreatitis (AP) was limited. Furthermore, it is important to note that our research is based on data obtained from a single database, which raises questions about the universality of our results in a wider population. In order to address the limitations found in our research, future research should consider prospective study design, multicenter collaboration, and larger datasets to establish causal relationships, enhance external validity, and promote research results to different populations.

**Table 3. The association between two groups of anion gap and acute kidney injury.**

| Variable | Crude Model | | Adjusted Model 1 | | Adjusted Model 2 | | Adjusted Model 3 | |
|---|---|---|---|---|---|---|---|---|
| | OR (95%CI) | P | OR (95%CI) | P | OR (95%CI) | P | OR (95%CI) | P |
| AG,mmol/L | 1.06 (1.03 ~ 1.1) | 0.001 | 1.06 (1.03 ~ 1.1) | 0.001 | 1.05 (1.02 ~ 1.09) | 0.005 | 1.05 (1.01 ~ 1.09) | 0.015 |
| AG<16mmol/L | 1(Ref) | | 1(Ref) | | 1(Ref) | | 1(Ref) | |
| AG≥16mmol/L | 2.35(1.57 ~ 3.53) | <0.001 | 2.41 (1.60 ~ 3.64) | <0.001 | 2.26(1.48 ~ 3.45) | 0.001 | 2.09 (1.35 ~ 3.24) | 0.001 |

Crude Model: no adjustment.

Adjusted Model 1: Adjusted for age, gender, and race.

Adjusted Model 2: Adjusted for model 1 plus heart rate, systolic blood pressure,diastolic blood pressure and blood oxygen degree of saturation.

Adjusted Model 3: Adjusted for model 2 plus blood lactic acid and sofas core.

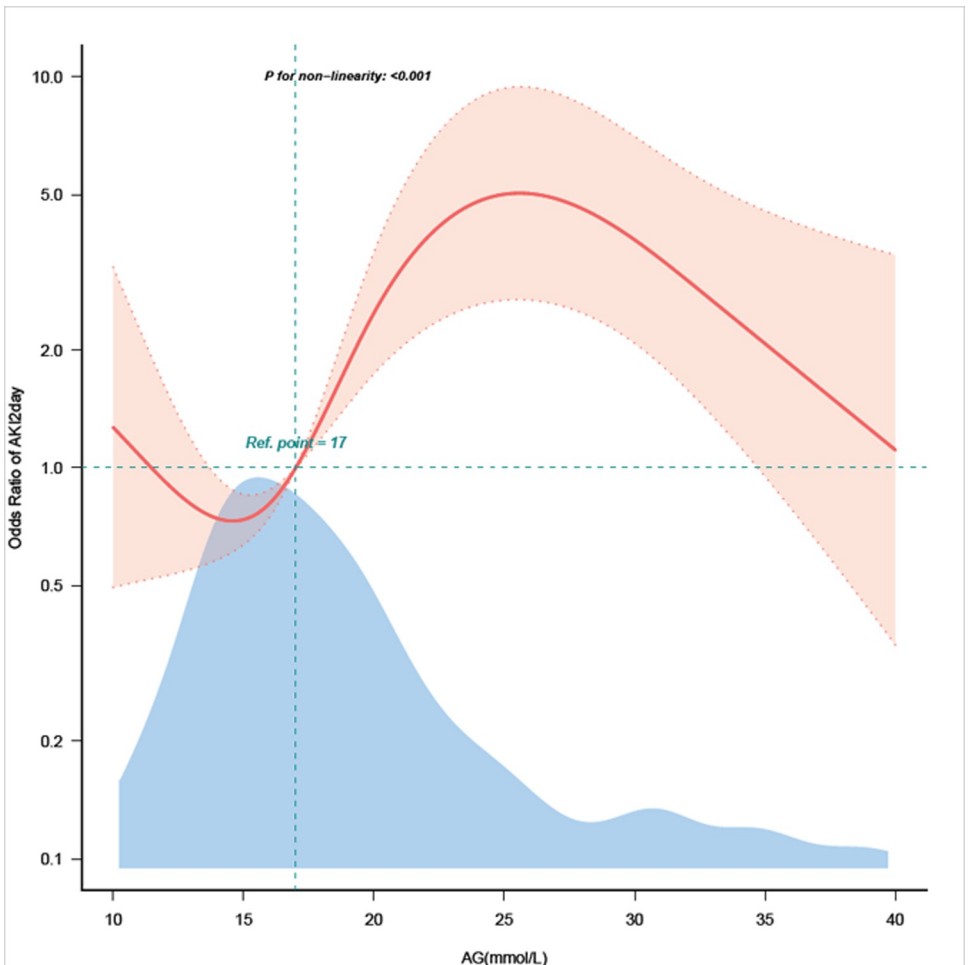

**Fig 2. Nonlinear correlation between serum anion gap and acute kidney injury.**

## Conclusion

Our study provides evidence for an independent correlation between elevated baseline anion gap (AG) levels and increased risk of acute kidney injury (AKI) in patients with acute pancreatitis (AP). The observed nonlinear S-shaped relationship underscores the importance of considering a specific AG when assessing AKI risk. Incorporating AG measurement into routine clinical practice has the potential to enhance risk assessment, early detection, and targeted management strategies for AKI in patients with AP. These findings highlight the clinical significance of AG as a potential biomarker of AKI in AP and underscore the importance of a personalized approach in risk assessment and patient management.Continued research in this area will help refine risk assessment models and develop targeted interventions to prevent and manage AKI in AP.

## Strengths and limitations

First, we are a large retrospective cohort study. Secondly, in order to mitigate potential confounding factors, we conducted multiple covariate adjustment and multiple model adjustment. As a retrospective study, we were limited in our ability to establish a causal relationship between baseline anion gap (AG) and acute kidney injury (AKI) in patients with acute

pancreatitis (AP). In addition, the data in this study were obtained from a single database, which may reduce the applicability of the results to other populations.

## Supporting information

**S1 Table. Relationship between inflection point and AKI.**
(DOCX)

**S1 File.**
(ZIP)

## Acknowledgments

We would like to extend our heartfelt thanks to Dr. Jie Liu, Department of Vascular Surgery, PLA General Hospital, for contributing to design consultation and comments on the manuscript; I would like to thank Dr. Qilin Yang of the Department of Critical Care Medicine of the Second Affiliated Hospital of Guangzhou Medical University for his guidance and contribution to the application and data extraction of MIMIC database.

## Author Contributions

**Data curation:** Rui Wang, Li Li, Qingwei Kong.

**Methodology:** Haiping Ma.

**Writing – original draft:** Yaqing Zhang.

**Writing – review & editing:** Cuiping Hao, Ying Zhang, Jinfeng Li.

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
