## [Decision Letter · Decision Letter 0]

23 May 2024

PONE-D-23-43806Relationship between serum anion gap and acute kidney injury in patients with acute pancreatitis upon admission: a retrospective study based on MIMIC-IV databasePLOS ONE

Dear Dr. li,

Thank you for submitting your manuscript to PLOS ONE. After careful consideration, we feel that it has merit but does not fully meet PLOS ONE’s publication criteria as it currently stands. Therefore, we invite you to submit a revised version of the manuscript that addresses the points raised during the review process.

We look forward to receiving your revised manuscript.

Kind regards,

Keiko Hosohata, Ph.D.

Academic Editor

PLOS ONE

Journal Requirements:

Reviewers' comments:

Reviewer's Responses to Questions

**Comments to the Author**

1. Is the manuscript technically sound, and do the data support the conclusions?

Reviewer #1: Partly

Reviewer #2: Partly

2. Has the statistical analysis been performed appropriately and rigorously? 

Reviewer #1: No

Reviewer #2: No

3. Have the authors made all data underlying the findings in their manuscript fully available?

Reviewer #1: No

Reviewer #2: Yes

4. Is the manuscript presented in an intelligible fashion and written in standard English?

Reviewer #1: No

Reviewer #2: Yes

5. Review Comments to the Author

Reviewer #1: In this retrospective study conducted in 400 critically ill patients admitted to the ICU for severe acute pancreatitis (AP), whose data were extracted from a large institutional database, the Author investigated the association of serum anion gap (SAG) with the development of acute kidney injury (AKI) within 48 hours since admission. After adjusting for several potential confounders, they found that SAG had a nonlinear association with the odds of AKI, and that SAG values > 16 mmol/L were associated with significantly higher odds of AKI compared with patients having SAG values < 16 mmol/L. They conclude that SAG is a useful biomarker that could be used to identify patients with AP at highest risk of developing AKI, and that it could also be used to monitor the progression of AKI in these patients.

I have several comments:

1) Because SAG had clearly a nonlinear association with the odds of AKI, patients should have been partitioned into SAG tertiles rather than using an arbitrary cut-off point at 16 mmol/L. The association with AKI could have been explored at multivariable analysis taking the lowest SAG tertile as a reference

2) Albumin-corrected SAG should have been used for all analyses, as it is conceivable that most of the enrolled patients, especially those with most severe clinical conditions at ICU admission, had low serum albumin levels. Hence SAG may have been underestimated in a non-uniform fashion in this population

3) In the same line of reasoning, because the degree of metabolic acidosis was more severe in more severe patients, additional adjustment for plasma bicarbonate is also advisable (see for instance ref #32 in the manuscript)

4) I believe that, because this study was conducted in critically ill patients, adjustment for SAPS II score at admission would have been preferable to adjusting for chronic comorbidities

5) As maximum SAG recorded within 24 hours since ICU admission was taken as the exposure measure in this study, an important limitation that should be acknowledged is that the effect of different volume and type (i.e., different chloride concentration) of resuscitation fluids administered within 24 hours, which represents an important confounding factor

6) In the Introduction section (page 5, lines 70-72) the Authors state “Prior research has shown a potential correlation between elevated AG levels and AKI in certain medical contexts, including sepsis and diabetic ketoacidosis”. This sentence is not strictly relevant to the aim of the study, and should be deleted

7) In the Discussion section (pag. 12, lines 240-244, the Authors state “Continuous AG measurement can provide valuable information on the progression of AKI, helping to intervene in a timely manner and develop personalized treatment plans. Secondly, incorporating AG measurement into the existing AKI prediction model maybe improve its accuracy and predictive ability in predicting AP patients”. These conclusion are not supported by the results, and should be deleted. The conclusions must be more nuanced, especially given the fact that the retrospective design of the study prevents establishing causation.

8) The English language throughout the text of the manuscript should be extensively edited by a native English-speaking person

Minor points

1) The term “association” should be used in lieu of “correlation” throughout the text

2) In the Methods section (page 6, line 97) what do the Authors mean by “not having sufficient data on AP”? Please explain in more details

3) In the Statistics section (page 7, line 130) the Authors mention that “several interpolation methods” were employed for variables with <10% missing data. Please, explain in more details which were the specific interpolation techniques that were used

4) Page 7, line 133: please change “logical” to “logistic”

Reviewer #2: Relationship between serum anion gap and acute kidney injury in patients with acute pancreatitis upon admission: a retrospective study based on MIMIC-IV database

Review

1. The authors herein presented a good paper shedding light on the association between the old age indicator for acute kidney injury (AKI) and serum anion gap (AP) and its correlation in acute pancreatitis (AP). The authors conclude that AG has a correlation with the incidence of AKI in AP patients admitted to the clinics within 48 hours.

2. Inasmuch as the authors have made a significant point that is worth considering in clinical practice in the monitoring of the risk of AP to the development of AKI, albeit my recommendations for improvement.

3. Firstly, the title could be revised to accurately reflect the time point of the investigation and the application of AG in AP-induced AKI. From the analysis, AG values were considered relevant only after 48 hours after admission, which suggests that AG may be relevant only at this time point. This is because AG is not a static indicator and is highly subject to alterations due to medications or physiological or pathological conditions. Moreover, the first use of the abbreviation MIMIC-IV sends any reader to head-scratching, as one may not know what it is. It may be that the use of a single center study or the site of the study could be appropriate

4. Secondly, the abstract did not point to the real conclusion and value of the study apart from informing readers on the rationale of the study. It is important that abstracts tell the story and its conclusion to attract interest from readers.

5. Thirdly, although the authors focussed on AG, it would have been nicer and more convincing to attempt to explain step wisely the risk factors to AP and AKI, then later situate in the AG values relative to the AP clinical indicators and AKI measures of sCr and other markers of AKI. Of course, the authors have presented all in Table 1 that is sometimes cumbersome for readers.

6. Lastly, the authors should re-check the grammar and referencing as some statements are made without appropriate referencing.

6. PLOS authors have the option to publish the peer review history of their article (what does this mean?). If published, this will include your full peer review and any attached files.

Reviewer #1: No

Reviewer #2: **Yes: **Adu Gyamfi Michael

---

## [Author Response · Author response to Decision Letter 0]

11 Jul 2024

Responds to the reviewer’s comments:

Reply to Reviewer #1

Dear Reviewer,

Thank you very much for your valuable time to review the manuscript, as well as your encouraging comments on its merits. We have made peer-to-peer revisions based on your comments, which we hope you can accept.

Comments: 

In this retrospective study conducted in 400 critically ill patients admitted to the ICU for severe acute pancreatitis (AP), whose data were extracted from a large institutional database, the Author investigated the association of serum anion gap (SAG) with the development of acute kidney injury (AKI) within 48 hours since admission. After adjusting for several potential confounders, they found that SAG had a nonlinear association with the odds of AKI, and that SAG values > 16 mmol/L were associated with significantly higher odds of AKI compared with patients having SAG values < 16 mmol/L. They conclude that SAG is a useful biomarker that could be used to identify patients with AP at highest risk of developing AKI, and that it could also be used to monitor the progression of AKI in these patients.I have several comments:

1) Because SAG had clearly a nonlinear association with the odds of AKI, patients should have been partitioned into SAG tertiles rather than using an arbitrary cut-off point at 16 mmol/L. The association with AKI could have been explored at multivariable analysis taking the lowest SAG tertile as a reference.

2) Albumin-corrected SAG should have been used for all analyses, as it is conceivable that most of the enrolled patients, especially those with most severe clinical conditions at ICU admission, had low serum albumin levels. Hence SAG may have been underestimated in a non-uniform fashion in this population.

3) In the same line of reasoning, because the degree of metabolic acidosis was more severe in more severe patients, additional adjustment for plasma bicarbonate is also advisable (see for instance ref #32 in the manuscript).

4) I believe that, because this study was conducted in critically ill patients, adjustment for SAPS II score at admission would have been preferable to adjusting for chronic comorbidities

5) As maximum SAG recorded within 24 hours since ICU admission was taken as the exposure measure in this study, an important limitation that should be acknowledged is that the effect of different volume and type (i.e., different chloride concentration) of resuscitation fluids administered within 24 hours, which represents an important confounding factor.

6) In the Introduction section (page 5, lines 70-72) the Authors state “Prior research has shown a potential correlation between elevated AG levels and AKI in certain medical contexts, including sepsis and diabetic ketoacidosis”. This sentence is not strictly relevant to the aim of the study, and should be deleted.

7) In the Discussion section (pag. 12, lines 240-244, the Authors state “Continuous AG measurement can provide valuable information on the progression of AKI, helping to intervene in a timely manner and develop personalized treatment plans. Secondly, incorporating AG measurement into the existing AKI prediction model maybe improve its accuracy and predictive ability in predicting AP patients”. These conclusion are not supported by the results, and should be deleted. The conclusions must be more nuanced, especially given the fact that the retrospective design of the study prevents establishing causation.

The English language throughout the text of the manuscript should be extensively edited by a native English-speaking person.

Minor points

1) The term “association” should be used in lieu of “correlation” throughout the text.

2) In the Methods section (page 6, line 97) what do the Authors mean by “not having sufficient data on AP”? Please explain in more details.

3) In the Statistics section (page 7, line 130) the Authors mention that “several interpolation methods” were employed for variables with <10% missing data. Please, explain in more details which were the specific interpolation techniques that were used.

4) Page 7, line 133: please change “logical” to “logistic”.

Comment 1:

Because SAG had clearly a nonlinear association with the odds of AKI, patients should have been partitioned into SAG tertiles rather than using an arbitrary cut-off point at 16 mmol/L. The association with AKI could have been explored at multivariable analysis taking the lowest SAG tertile as a reference.

Response 1:

Thank you for your suggestion. The normal value of anion gap (AG) is 8-16mmol/L. For clinical purposes, AG is grouped according to the cut-off value of 16mmol/L. However, according to your suggestion, we also analyzed the data of AG according to the three quantile, and the cut-off values of the quantile were 14mmol/L and 17mmol/L. The results of multivariate analysis showed that compared with the < 14mmol/L group, the 14-17mmol/L group had no clinical significance (OR=1.51, 95%CI:0.95 to 2.41, p-value=0.082), while the>17mmol/L group had a significantly increased risk of AKI (OR=3.77,95%CI:2.26-6.27, p-value<0.001). The results were consistent with those of the high anion gap (≥16mmol/L) group (OR=2.35,95%CI: 1.57 to 3.53, p-value<0.001). So we'll stick with the original grouping. The results of the modified two-group and three-digit multi-factor analysis are available in the document and supporting table.(Table3,S1table)

Comment 2:

Albumin-corrected SAG should have been used for all analyses, as it is conceivable that most of the enrolled patients, especially those with most severe clinical conditions at ICU admission, had low serum albumin levels. Hence SAG may have been underestimated in a non-uniform fashion in this population.

Response 2:

This is a very good proposal. We have reorganized the data, calculated and added the corrected albumin anion gap as a covariate according to the formula(S2 Table). We have also analyzed the relationship between the corrected albumin anion gap and the prognosis of patients with acute pancreatitis who develop AKI(S1 Fig), and the results are also included in the appendix, which will be another research direction for us in the future.

Comment 3:

In the same line of reasoning, because the degree of metabolic acidosis was more severe in more severe patients, additional adjustment for plasma bicarbonate is also advisable (see for instance ref #32 in the manuscript).

Response 3:

Thanks for your suggestion, we also made more model adjustments in the multivariate analysis to further validate the stability of the results, model 4 has added to the adjustment of bicarbonate covariates,model 5 has added to the adjustment of albumin, model 6 has added to the adjustment of both albumin and bicarbonate covariates concurrently, the results shows that remained stable(S4 Table).

Comment 4:

I believe that, because this study was conducted in critically ill patients, adjustment for SAPS II score at admission would have been preferable to adjusting for chronic comorbidities.

Response 4:

Thank you for your comments. According to your suggestion, we updated the adjustment of covariate and added the critical illness score. Yue S, et al.[1] showed that SOFA score and SapsII score were basically consistent in the accuracy of predicting AKI in critically ill patients. Therefore, in the multifactor regression analysis, I chose to add adjustments for SOFA scores (Model 3), which is consistent with the initial results.

Comment 5:

As maximum SAG recorded within 24 hours since ICU admission was taken as the exposure measure in this study, an important limitation that should be acknowledged is that the effect of different volume and type (i.e., different chloride concentration) of resuscitation fluids administered within 24 hours, which represents an important confounding factor.

Response 5:

In this study, we obtained the baseline AG values within 24 of acute pancreatitis patients admitted to the ICU, without including the fluid type of volume resuscitation, but this is a very good proposal and will be our research direction in the future.

Comment 6:

In the Introduction section (page 5, lines 70-72) the Authors state “Prior research has shown a potential correlation between elevated AG levels and AKI in certain medical contexts, including sepsis and diabetic ketoacidosis”. This sentence is not strictly relevant to the aim of the study, and should be deleted.

Response 6:

We are very sorry for our negligence that this sentence has been deleted and changed in the manuscript.

Comment 7:

In the Discussion section (pag. 12, lines 240-244, the Authors state “Continuous AG measurement can provide valuable information on the progression of AKI, helping to intervene in a timely manner and develop personalized treatment plans. Secondly, incorporating AG measurement into the existing AKI prediction model maybe improve its accuracy and predictive ability in predicting AP patients”. These conclusion are not supported by the results, and should be deleted. The conclusions must be more nuanced, especially given the fact that the retrospective design of the study prevents establishing causation.

The English language throughout the text of the manuscript should be extensively edited by a native English-speaking person.

Response 7:

Your proposal is very good, and this content will continue to be presented in our future research series, which has been deleted in the revised manuscript.

However, we do invite a friend of us who is a native English speaker to help polish our article. And we hope the revised manuscript could be acceptable for you.

In addition, we have also revised the following details that you have proposed:

Comment Minor points 1):

The term “association” should be used in lieu of “correlation” throughout the text.

Response 1):

Thank you for reviewing the quality of the manuscript and replacing "association" with "correlation" in revising the manuscript.

Comment Minor points 2):

In the Methods section (page 6, line 97) what do the Authors mean by “not having sufficient data on AP”? Please explain in more details.

Response 2):

Thank you for your questions. This is one of the exclusion criteria in the study population scheduling process, which refers to the population of patients with pancreatitis with missing key data, such as lactate and albumin. According to your above opinions, we have reorganized the data inclusion and exclusion process and added covariates. The specific amount of missing data has been presented in the latest flow chart(Fig1).

Comment Minor points 3):

In the Statistics section (page 7, line 130) the Authors mention that “several interpolation methods” were employed for variables with <10% missing data. Please, explain in more details which were the specific interpolation techniques that were used.

Response 3):

Thank you again for your positive suggestions. In the new data, in order to prevent confounding factors, we directly deleted the data with covariate missing≥10%, and the variables with<10% missing we used multiple imputation, based on 5 replications method in the Free statistics program to explain the missing data of the covariate using multiple interpolation based on 5 repeats. We have carefully rephrased the statements in the revised draft and used recent references.

Comment Minor points 4):

Page 7, line 133: please change “logical” to “logistic”.

Response 4):

We are very sorry for our writing negligence, the revised draft has been change "logical" to "logistic".

Reply to Reviewer #2

Dear Reviewer,

Thank you again for your positive comments and valuable suggestions on improving the quality of your manuscript. According to the suggestions, we have made serious revisions and hope to get your approval.

Reply to Reviewer #2

Dear Reviewer,

Thank you again for your positive comments and valuable suggestions on improving the quality of your manuscript. According to the suggestions, we have made serious revisions and hope to get your approval.

Comments 1:

The authors herein presented a good paper shedding light on the association between the old age indicator for acute kidney injury (AKI) and serum anion gap (AP) and its correlation in acute pancreatitis (AP). The authors conclude that AG has a correlation with the incidence of AKI in AP patients admitted to the clinics within 48 hours.

Response 1:

Thank you very much for your approval and high quality evaluation of the manuscript. We will definitely revise it one to one and submit a better manuscript according to your suggestions.

Comments 2:

Inasmuch as the authors have made a significant point that is worth considering in clinical practice in the monitoring of the risk of AP to the development of AKI, albeit my recommendations for improvement.

Response 2:

Although your proposal is great, we are a retrospective cohort study predicting AKI risk of AP will continue in our future research, which will also be a motivation for our work.

Comments 3:

Firstly, the title could be revised to accurately reflect the time point of the investigation and the application of AG in AP-induced AKI. From the analysis, AG values were considered relevant only after 48 hours after admission, which suggests that AG may be relevant only at this time point. This is because AG is not a static indicator and is highly subject to alterations due to medications or physiological or pathological conditions. Moreover, the first use of the abbreviation MIMIC-IV sends any reader to head-scratching, as one may not know what it is. It may be that the use of a single center study or the site of the study could be appropriate.

Response 3:

Thank you very much for your constructive comments, our data extracted the baseline AG value from the first ICU entry, we have modified the title to express it more accurate, and explained the MIMIC-IV database abbreviation when it first appeared in the article. The latest article is entitled "Correlation between baseline anion gap and early acute kidney injury in patients with acute pancreatitis in the intensive care unit: a single-center retrospective cohort study".

Comments 4:

Secondly, the abstract did not point to the real conclusion and value of the study apart from informing readers on the rationale of the study. It is important that abstracts tell the story and its conclusion to attract interest from readers.

Response 4:

Thank you for your comments. I am sorry to say that we did not modify the primary title and secondary title in accordance with the journal format, so that we wrote the conclusions of the study in the abstract and the value was not seen. In the revised draft, we have carefully modified the format and improved the content of the abstract.

Comments 5:

Thirdly, although the authors focussed on AG, it would have been nicer and more convincing to attempt to explain step wisely the risk factors to AP and AKI, then later situate in the AG values relative to the AP clinical indicators and AKI measures of sCr and other markers of AKI. Of course, the authors have presented all in Table 1 that is sometimes cumbersome for readers.

Response 5:

Thank you for your comments. We mentioned in the Materials and methods that the diagnosis of AKI is defined by the level of serum creatinine (SCr), which does not conflict with the inclusion of blood creatinine in the covariable.

Comments 6:

Lastly, the authors should re-check the grammar and referencing as some statements are made without appropriate referencing.

Response 6:

Thank you for your suggestions on the quality of the article. We have reviewed and checked the grammar and references of the whole article, and all changes have been marked in the revised version.

---

## [Decision Letter · Decision Letter 1]

15 Nov 2024

PONE-D-23-43806R1Correlation between baseline anion gap and early acute kidney injury in patients with acute pancreatitis in the intensive care unit: a single-center retrospective cohort studyPLOS ONE

Dear Dr. li,

Thank you for submitting your manuscript to PLOS ONE. After careful consideration, we feel that it has merit but does not fully meet PLOS ONE’s publication criteria as it currently stands. Therefore, we invite you to submit a revised version of the manuscript that addresses the points raised during the review process.

We look forward to receiving your revised manuscript.

Kind regards,

Keiko Hosohata, Ph.D.

Academic Editor

PLOS ONE

Reviewers' comments:

Reviewer's Responses to Questions

**Comments to the Author**

1. If the authors have adequately addressed your comments raised in a previous round of review and you feel that this manuscript is now acceptable for publication, you may indicate that here to bypass the “Comments to the Author” section, enter your conflict of interest statement in the “Confidential to Editor” section, and submit your "Accept" recommendation.

Reviewer #3: All comments have been addressed

Reviewer #4: (No Response)

2. Is the manuscript technically sound, and do the data support the conclusions?

Reviewer #3: Yes

Reviewer #4: Yes

3. Has the statistical analysis been performed appropriately and rigorously? 

Reviewer #3: Yes

Reviewer #4: Yes

4. Have the authors made all data underlying the findings in their manuscript fully available?

Reviewer #3: Yes

Reviewer #4: Yes

5. Is the manuscript presented in an intelligible fashion and written in standard English?

Reviewer #3: Yes

Reviewer #4: Yes

6. Review Comments to the Author

**Reviewer #3: **The authors answered all the questions addressed. Congratulations to all authors for the research.

No more questions

**Reviewer #4:** 1. Grammar has improved from prior submission, but a few corrections are still needed in terms of grammar eg. check flow diagram.

2. Since this is an adult only data, authors do not need to include "excluded participants age <18 years n=0" in the flow diagram. Instead, a sentence should be added to the methods that the population consists of individuals aged 18 years or older.

7. PLOS authors have the option to publish the peer review history of their article (what does this mean?). If published, this will include your full peer review and any attached files.

Reviewer #3: **Yes: **Miguel Angelo Goes

Reviewer #4: No

---

## [Author Response · Author response to Decision Letter 1]

21 Nov 2024

Dear Reviewers, 

Thank you very much for your suggestion on the flow chart. Following your suggestion, we have conducted a further review and have removed the reference to "exclude participants aged <18 years n=0" and have instead included an explicit statement in the Methods section. This clearly illustrates that our study population consists only of individuals ≥18 years old, thus simplifying the flow chart and improving the clarity of our approach. We also ensured that the language was fluent and coherent to enhance readability. We believe that these additional corrections will meet your expectations, and we look forward to your evaluation of the revised flowchart.

---

## [Editor Report · Decision Letter 2]

26 Nov 2024

Correlation between baseline anion gap and early acute kidney injury in patients with acute pancreatitis in the intensive care unit: a single-center retrospective cohort study

PONE-D-23-43806R2

Dear Dr. li,

We’re pleased to inform you that your manuscript has been judged scientifically suitable for publication and will be formally accepted for publication once it meets all outstanding technical requirements.

Kind regards,

Keiko Hosohata, Ph.D.

Academic Editor

PLOS ONE
---

## [Editor Report · Acceptance letter]

12 Dec 2024

PONE-D-23-43806R2 

PLOS ONE

Dear Dr. Li, 

I'm pleased to inform you that your manuscript has been deemed suitable for publication in PLOS ONE. Congratulations! Your manuscript is now being handed over to our production team.

Kind regards, 

on behalf of

Dr Keiko Hosohata 

Academic Editor

PLOS ONE